# Adding Knowledge to Virtual Teams in the New Normal: From Leader-Team Communication towards the Satisfaction with Teamwork

**Elena-Mădălina Vătămănescu [1,\*]** , **Elena Dinu [2]** , **Mădălina-Elena Stratone [2]** , **Roxana-Maria Stăneiu [2]** and **Florina Vintilă [2]**

[1] Faculty of Management, National University of Political Studies and Public Administration (SNSPA), 30A Expozitiei Blvd., 012104 Bucharest, Romania

[2] Doctoral School in Management, National University of Political Studies and Public Administration (SNSPA), 30A Expozitiei Blvd., 012104 Bucharest, Romania; elena.dinu@facultateademanagement.ro (E.D.); madalina.stratone@facultateademanagement.ro (M.-E.S.); roxana.staneiu@gmail.com (R.-M.S.); florinavintila2@gmail.com (F.V.)

[\*] Correspondence: madalina.vatamanescu@facultateademanagement.ro; Tel.: +40-748-119-900

**Abstract:** The present paper sets out to investigate the relationships among several key constructs that cover the work patterns and processes in the context of the COVID-19 pandemic. Emphasis is laid on the leader-team communication, the fostering of a strong team culture, team performance and satisfaction with teamwork in the case of virtual teams. The scrutiny is intended to complement recent developments in the field which compared traditional and virtual teams at different levels by adding knowledge to virtual teams' communication and interaction patterns and processes. In this vein, an online survey was conducted with 175 members from different virtual teams. The findings showed the advancement of a pertinent conceptual model, mostly displaying significant relationships among constructs. Four out of the five formulated hypotheses were validated, the highest influences being reported between leader-team communication and team culture, respectively, and between team performance and satisfaction with teamwork. Furthermore, the structural model explained over 50% of the variance in the satisfaction with teamwork, thus supporting the relevance of the inferred relationships.

**Keywords:** virtual teams; team culture; leader-team communication; team performance; satisfaction with teamwork; new normal

## 1. Introduction

During the last two years, many companies from various industries have been fundamentally impacted by the COVID-19 pandemic. The COVID-19 crisis, as explained by Bratianu and Bejinaru [1] (p. 11), "came like any other natural disaster, finding people and organizations unprepared for disruptive power and social nexus". Business models, organizational interactions, the work climate and living patterns have dramatically changed [2,3]. The outspread of the virus has significantly altered the world of work, disrupting the ways organizations manage their businesses and especially how leaders engage with their teams.

Overall, the COVID-19 pandemic created the boundaries of a "new normal" that drastically impacted the working environment and how organizations reacted to the crisis situation, on both strategic and operational levels. The pandemic required the development of new managerial strategies to cope not only with the collective fear, but also with the challenges implied by a full migration to a virtual workplace with dispersed team members. From explaining the new reality and promoting a culture of trust, to upgrading communication tools and approaches to better convey information to employees, managers and leaders needed to adjust to respond to the crisis imperatives [4].

A critical issue right now, when most organizations plan to return their workforce to the physical workplaces, is to understand how organizations are coping with the new normal in terms of work dynamics and keeping employees engaged, and to what extent virtual teams may have proved beneficial in terms of team performance (i.e., task completion and objectives achievement) [5]. Overall, remote work has engendered multiple opportunities, but there are still many new challenges ahead. Teams have become more isolated and digital communication and interaction may raise unexpected issues at different levels (i.e., how team culture is formed and consolidated, how job satisfaction is affected, etc.).

Companies are reimagining work and establishing a new hybrid approach to working is meant to support their people today and in the future. A tool that could be used as a starting point in understanding this new hybrid workforce is the Microsoft 2021 Work Trend Index: Annual Report called "The next great disruption is hybrid work—are we ready?" The report outlines findings from a study of more than 30,000 people in 31 countries, focusing on issues like productivity and labor characteristics, and brought to the fore virtuality as a key medium for future teamwork. Managers and leaders have been challenged to develop emergent strategies to cope with the organizational and teamwork dynamics, to advance new plans and explore new opportunities for mitigating the disruptive effects on people and business [1] (p. 15). To support this idea, Deloitte [6] has shown in one of its studies that the social and economic crisis caused by the current pandemic is an extreme but relevant example of the types of challenges leaders face today. In times of uncertainty, there is a stronger call for a more human-centric leadership style.

In a review of the literature conducted by Moroșan-Dănilă, Grigoraș-Ichim and Bordeianu [7], challenges of virtual work (i.e., telework) have been extracted, which are reflected in varied technological, communication and leadership issues. They are interconnected and not paying attention to these areas can have negative effects, such as isolation and work disfunctions, as also highlighted by Bentley et al. [8], Eddleston and Mulki [9], and Wang, Liu, Qian and Parker [10]. In return, these can have a negative impact on wellbeing (physical and mental health) and work performance [11]. In consequence, companies must not forget that the most important resource they have at their disposal and one that they can always adapt to (with the right impulse) is the human resource [3,12,13]. Organizing virtual work teams was a necessity in 2020, but it has become a long-term solution to achieve the company's cost-cutting objectives and efficiency.

As the results from the systematic literature review conducted by Chafi et al. [14], the main positive effects of remote work are increased flexibility, autonomy, job satisfaction and a better work–life balance. On the contrary, the perverse effects encompass social isolation, uncertainty about professional development, an "always-on" culture and work overload. The next decade will be one of significant experimentation concerning work and there is still much to be learnt about effective approaches [1,5]. In a virtual-centric work environment, it will be critical for business leaders to understand their employees' dynamics and to invest in their personnel with a view to help them reach their full potential.

Building on the dialectic argumentation regarding the effects of virtual teamwork and on the topicality of a "moving target" research theme, the present paper sets out to investigate the relationships among several key constructs that cover the work patterns and processes in the context of the new normal. Emphasis is laid on the effectiveness of leader-team communication, the strength of the team culture and on achieving a higher level of team performance and of satisfaction with teamwork in the case of virtual teams. The scrutiny thus gives credit to recent developments in the field (e.g., [5]), which compared traditional and virtual teams at different levels and complemented previous results by extending the focus on and adding knowledge to virtual teams' communication and interaction processes. In addition, even though the exploration of virtual teams is not a novel topic per se, consistent with the observations of the United Nations [15], conducting such studies in the context of COVID-19 is more than opportune given that the pandemic has dramatically altered all socio-economic strata, leading to the loss of over 250 million full-time jobs. This situation has entailed the primacy of full and productive employment

and decent work for everybody as a paramount sustainable development goal. By revolving around the new patterns of working in the virtual environment and displaying key interaction processes among people—as a pivotal dimension of the Triple-Bottom-Line—the investigation contributes to a better understanding of the individuals' multifaceted accommodation to the new normal and of the potential antecedents of their wellbeing.

In this sense, a questionnaire-based survey was carried out between January and March 2022 with 175 respondents working in various virtual teams. The main condition for the selection of participants was for them to have worked in traditional teams before the pandemic so that the study capture the ratings of subjects that have been directly influenced by the work disruptions availed by COVID-19.

The paper was structured in four main sections following the introduction: the literature review and hypotheses formulation, material and methods, presentation of the results and the discussion and conclusions section.

## 2. Literature Review and Hypotheses Development

### 2.1. Communication between Leaders and Team Members and Team Performance

The effectiveness of the communication process between leaders and team members within virtual teams has been recently paid a lot of attention given that the clarity of task allocation, the accuracy of knowledge sharing and the interpersonal socio-emotional cues apposite for direct interaction have been dramatically challenged by the translation to the online work environment [1,5,7,10]. An extensive study of 80 software development teams of programmers from the United States, South America, Europe, and Asia has shown that virtual teams can lead to increased efficiency and better business results, but only if they are managed proactively via a good and effective communication process to maximize the potential benefits while minimizing the disadvantages [16]. The most important elements of success included establishing processes at the beginning of the task, communicating efficiently and keeping conflicts focused on the tasks. With a view to becoming an effective and high-performing team, especially when it comes to a virtual environment, some basic conditions of team performance must be reached, i.e., clearly defining tasks and objectives, the right skills, appropriate and developed roles, conflict management, performance management processes and effective communication [17,18].

A study conducted by Klus and Muller [19], including interviews with executives and an analysis of job advertisements for leadership positions, concluded that leadership skills in the context of disruptive technological transformations presume mostly the ability to communicate effectively through different channels, organization skills and the transmission of task-oriented knowledge. In addition, empathy and open-mindedness toward the team members are required. Other research developed by Klus and Muller [20] proposes that strong communication skills in a leadership role are necessary for knowledge sharing and ensuring task completion, as well as for spreading the word regarding digitalization-induced changes [21–23]. As digitalization redefines the communication channels and platforms, new skills and abilities should be developed to address the disruptions correspondingly [13,24–27].

Morrison-Smith and Ruiz [28] identified during an extensive literature review that team communication is a real challenge in the virtual environment (see also [29,30]). In addition, the perceptions of the members regarding how leaders capitalize on the means of communication influence the way they perceive team performance in general [31]. This implies that favorable evaluations of leader-team communication is conducive to favorable perceptions in what concerns team performance. Leaders have the ability to nourish trustworthy relationships in virtual teams if they leverage various communication tools and if they catalyze synchronous knowledge sharing. Unfolding constant leader-team communication flows entails task completion and commitment within the team [32].

While investigating how leaders use empathic skills during the pandemic to manage the wellbeing of their employees, so that they remained motivated and engaged for meaningful performance, Raina [33] concluded that consistent communication is the lynchpin

for the organizational success. Moreover, leaders who create a transparent, positive and participative communication environment have the ability to impact operations and achieve their objectives via increasing team performance. Similarly, Jawahar and Mohammed [34] confirmed within an empirical study that managers' process of self-efficacy significantly affects projects' completion, as good leaders are better at defining the objectives, describing and allocating the tasks, getting input, securing resources, monitoring task fulfilment, etc. Based on a systematic literature review, Cummings, Tate, Lee, Wong, Paananen, Micaroni and Chatterjee [35] have contended that leadership that is solely focused on task completion does not induce optimum results. Instead, relational leadership styles positively affect the labor force and organizational outcomes. By corroborating the main common facets of an effective communication process between leader and team members as inferred by the aforementioned studies [5,16,17,28,30,31], emphasis will be laid on the effectiveness of task communication via different channels, knowledge sharing for clarifying misunderstandings and the advancement of empathic communication. Thus, following the arguments brought forward by the literature review, H1 is proposed below:

**Hypothesis 1 (H1):** *The effectiveness of the communication between leaders and team members positively influences the level of team performance within virtual teams in the context of COVID-19 pandemic.*

*2.2. Communication between Leaders and Team Members and Team Culture*

Communication has been highlighted as an important pillar in the creation of a team culture for decades, being perceived, among leadership, social interaction and interdependence, as a contributor to the development of subculture in work units [36]. It facilitates relationship-building and trust among individuals [37], thus emerging as one of the most effective factors in establishing group cohesion [38].

When revolving around virtual teams, communication remains a point of reference, possessing a strong positive influence on team member's participation and engagement, elements that are inherent to the team culture [39]. In the virtual environment, leaders are challenged to focus on fostering close relationships with their team members to ensure engagement and a team climate of trust [40]. Virtual teams face various challenges on a daily basis, but communication between members stands out as one of the most striking factors [29]. Thus, a manager or a leader needs to adjust his way of communication, taking into account the team particularities and patterns of working [18,41].

When a leader develops a strong relation with the team members that is built on egalitarianism, authentic interaction and communication, there will be a strong positive effect on the team morale and spirit, which will generate important implications in organization commitment and employee loyalty as well [42]. A manager or a leader who is considered a good communicator acts accordingly within the team by promoting open dialogue, which allows the team members to share issues and concerns, values and norms, aspects that then positively contribute to the team culture creation [43–45]. A strong team culture may be also characterized by a high level of team members' engagement, and communication was found to play an essential role in this regard [46–49]. Effective communication garners trust among individuals, which has been advanced as a key prerequisite when it comes to building a healthy and strong team culture [50]. Effective communication promoted by the leaders may imply a positive effect on the team culture if it is supported by showing initiative, transparently exchanging information about team processes or giving and receiving feedback [51–54]. By conflating the extant research directions and giving credit to representative works in the field [36,37,50], within the scope of the current paper, the strength of the team culture was objectivized via the sharing the same values and goals, the co-creation of team climate based on trust and the presence of a strong team spirit. Based on the reviewed literature, hypothesis H2 is put forward:

**Hypothesis 2 (H2):** *The effectiveness of the communication between leaders and team members positively influences the strength of team culture within virtual teams in the context of the COVID-19 pandemic.*

*2.3. Team Culture and Team Performance*

Team culture has been highlighted as a strong incentive when it comes to how fast team members complete their tasks and achieve their objectives. Team culture can build a common representation of the work that needs to be completed, which allows members to prioritize together what is important for reaching the desired outcomes [55]. González-Romá, Fortes-Ferreira and Peiró [56] analyze the impact of team climate and culture on team performance from a four-dimension point of view, taking into account support from the organization, innovation, goal achievement and enabling formalization, looking at performance from both team members' and managers' perspective. They reached the conclusion that support from the organization, which is an important element for the entire organizational culture per se, had a positive influence on the performance perceived by both team members and managers [56]. Organization support promotes the creation of a strong team culture, which has a positive impact on individual and team performance [57,58].

A strong team culture positively influences the feeling of trust among its members, which then ensures a boosted knowledge-sharing appetite [59], leading to a higher team performance. Team culture offers the team members the right context to feel safe and drive innovation, elements that pave the road to an enhanced level of team performance. By employing a quantitative research approach based on the responses of 535 people from 95 teams, Jamshed and Majeed [60] demonstrated a positive relation between the strength of the team culture and team performance through knowledge sharing.

In a recent study, de Castro [61] looks into the relationship between a strong team culture and team performance and concludes that team culture has a positive influence on team performance, via the sharing of a common set of norms, values, beliefs and insights. In addition, it has been found that the higher the team commitment, the greater its performance insofar as people from committed teams believe in the same values, show interest in the projects' success and care about their work [25,49,61]. Furthermore, in the context of COVID-19, a culture based on trust, empowerment and cohesion was identified as a significant and positive determinant of virtual teams' performance [62]. Based on the main theoretical underpinnings regarding team performance [44,49,56–58] within the scope of the current research endeavor, the construct was measured via three pillars, namely the effectiveness in meeting the established objectives, finishing tasks on time and good coordination resulting in the usage of less resources. Hence, hypothesis H3 follows suit to the research presented above and proposes:

**Hypothesis 3 (H3):** *The strength of team culture positively influences the level of team performance within virtual teams in the context of the COVID-19 pandemic.*

*2.4. Team Performance and Satisfaction with Teamwork*

Regularly, job satisfaction describes how people feel about their work, what they expect from it, how they interact with the other team members and the leaders and what opportunities are offered to them to evolve and develop new skills and competences [63]. While many psychological, behavioral, social, economic, cultural and other factors determine a person's satisfaction with their job [64], Aziri [65] underlined the causal relation between satisfaction and one's work-related sense of achievement and success, as well as personal well-being. The author's review of the literature on satisfaction revealed that various factors such as motivation, attitude, expectations, work conditions, leadership, performance and rewards can all prove relevant for a person's commitment to an organization and their perception of fulfilment. Nevertheless, a straightforward causal relationship between job satisfaction and performance has not been proven yet.

Hodson [66] analyzed various workplace ethnographies and confirmed the importance of the pride derived by workers from task completion and its influence on work satisfaction. On the contrary, failure to comply with the job requirements induced unpleasant feelings in workers. Furthermore, pride in task completion significantly influences the organizational citizenship behavior, expressed through voluntary extra time afforded to the task, self-monitoring of collective results, on-the-job peer training and cooperation. Liu, Chen, Kin and Li [67] have established that, while the affective commitment of workers does not predict task performance, its relationship with work satisfaction is significantly moderated by the job completion.

In more comprehensive research, Park and Lee [68] have conducted a study on the relationship of work satisfaction with task satisfaction, organizational commitment and turnover intention, and found that workers scored payment lower than co-workers as an important factor for the work environment. Furthermore, the most relevant causal element that impacted satisfaction with the performed work, organizational commitment and turnover intention was job achievement. Similar findings have been put forward in a study by Ineson and Berechet [69] concerning satisfaction factors for employees of Romanian hotels. It has been concluded that staff satisfaction with their work is significantly influenced by the accomplishment of work objectives and performance.

In their qualitative study, Fitzgerald, Yates, Benger and Harris [70] concluded based on the interview results that working under systemic pressure in an uncontrollable environment led to diminished work satisfaction and well-being, as staff was feeling under-valued in their task achievement. Wang, Liu, Qian and Parker [10] have identified work–home interference, ineffective communication, procrastination and loneliness as key challenges for remote work, and these can be affected by work characteristics such as social support, job autonomy, monitoring and workload, and moderated by the workers' self-discipline, all of them impacting job performance and well-being. In addition, Orhan, Rijsman and van Dijk [71] have established that it is rather task virtuality and not team virtuality that significantly impacts workplace isolation, satisfaction, perceived performance and turnover intention in organizations. Virtuality in this respect is determined by a lack of physical interaction, but also interdependency between tasks or team members in completing their work.

As can be noticed from the literature review, there is very little research that straightforwardly empirically studied the relationship between satisfaction and team performance within virtual teams. In this vein, given that the empirical investigation was intended to approach only subjects who have moved to the virtual work environment after the outbreak of the pandemic, the satisfaction with teamwork has focused on two key dimensions, that is, enjoying teamwork in virtual teams and openness to support teamwork in the virtual environment, as two pivotal aspects indicative of how they feel about the new working conditions. By doing so, the paper aimed at enriching the knowledge on the team members' ratings of virtual interactions with new insights and advanced the following hypothesis:

**Hypothesis 4 (H4):** *The level of team performance positively influences the level of satisfaction with teamwork within virtual teams in the context of the COVID-19 pandemic.*

### 2.5. Team Culture and Satisfaction with Teamwork

Via the sharing of values, perceptions and beliefs and reconciling differences among peers [72–74], the organizational culture ensures internal integration and further satisfaction with teamwork [75]. A strong culture has the power to help co-workers accomplish easier their tasks and goals and become satisfied with the performed activities [76,77]. In this front, team culture comes forward as an important pillar of a team member's satisfaction, as it fosters higher engagement and catalyzes better communication and stronger social interaction [78].

The literature approaching the topic of team culture and satisfaction within the virtual teams is still developing, thus extending the knowledge about their benefits and

challenges [5,79]. According to different studies, work satisfaction is influenced by organizational culture at a macro level and by team culture at a micro level [80], depending on how members of those organizations/teams accept the already existing culture when they start a new job [81], on how their moral and physical needs are meet [82], on how culture is shaped over time, on how motivation is stimulated [83,84] and on how the organization is structured. In most cases, members who perceive a strong team culture have a greater sense of satisfaction when it comes to their jobs [44,49,85]. Based on the aforementioned considerations, hypothesis H5 is advanced:

**Hypothesis 5 (H5):** *The strength of team culture positively influences the level of satisfaction with teamwork within virtual teams in the context of the COVID-19 pandemic.*

Corroborating all the inferred relationships, the following research model was proposed (see Figure 1):

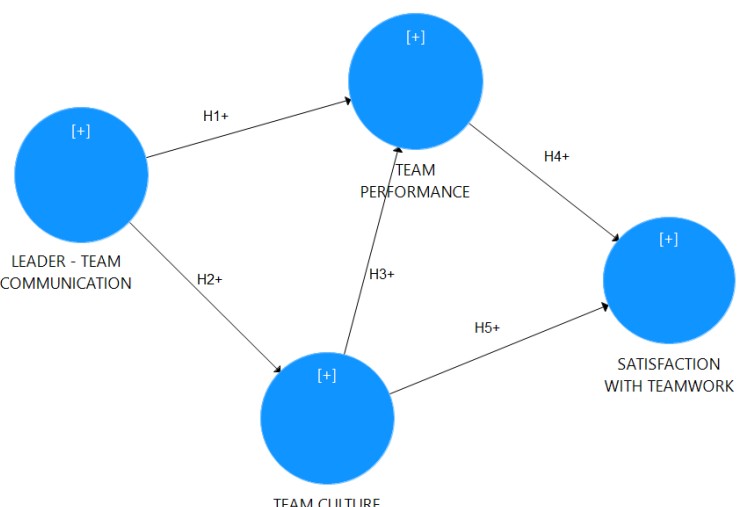

**Figure 1.** Conceptual model.

## 3. Materials and Method

### 3.1. Sample and Data Collection

A questionnaire-based survey was carried out between January and March 2022 with 175 respondents working in various virtual teams. Invitations were sent via email to more than 300 key informants; therefore, the response rate was around 58%. A snowball sampling technique was applied to reach potential participants in the study who complied with the criterion of working in virtual teams. Consequently, almost 25% of the completed questionnaires were filled in by members recommended by primarily contacted respondents. The sample characteristics are illustrated in Table 1.

With a view to properly estimate the minimum sample size, a G*Power Analysis was performed in line with Cunningham and McCrum-Gardner [86]. The results of the computation indicate that for an $f^2$ size effect of 0.15, a minimum sample of 89 participants would be necessary. Consequently, the reliability of the statistical analysis is validated given that the present research sample (e.g., 175) exceeds the minimum required by G*Power Analysis.

**Table 1.** Sample characteristics.

|  | Frequency | Percentage |
|---|---|---|
| No. of participants | 175 | 100% |
| Gender | | |
| Female | 110 | 62.9% |
| Male | 65 | 37.1% |
| Age | | |
| 18–24 | 52 | 29.8% |
| 25–34 | 64 | 36.6% |
| 35–44 | 43 | 24.5% |
| >45 | 16 | 9.1% |
| Organization type | | |
| Small and medium-sized enterprises | 51 | 29.1% |
| Multinational corporation | 73 | 41.7% |
| Public organizations | 36 | 20.6% |
| Other | 15 | 8.6% |
| Work experience (years) | | |
| 0–1 | 5 | 2.9% |
| 1–3 | 31 | 17.8% |
| 3–5 | 37 | 21.1% |
| 5–10 | 30 | 17.1% |
| >10 | 72 | 41.1% |

### 3.2. Procedure and Method

The aim of the empirical study is to explore the relationships among the effectiveness of the communication between leaders and team members, the strength of team culture, the level of team performance and the level of satisfaction with teamwork. Starting from the exploratory nature of the undertaking, the investigation resorts to the usage of partial least squares structural equation modeling (PLS-SEM) via SmartPLS 3.0 software [87]. In this sense, the analysis of the measurement and structural models is conducted by taking into consideration the rules of thumb and recommended guidelines for PLS-SEM applicability, as detailed in the following sections.

### 3.3. Measures

The research instrument is elaborated based on the reviewed literature, each item in the questionnaire being rated on a five-point Likert scale (1 = to a very small extent and 5 = to a very great extent). Four constructs are proposed, namely effectiveness of communication between leaders and team members (reflective construct comprising three items), strength of team culture (reflective construct comprising three items), level of team performance (reflective construct comprising three items) and level of satisfaction with teamwork (reflective construct comprising two items). The questionnaire initially included five items (i.e., indicators) for each construct, but after performing the measurement model assessment, the indicators that did not conform to the validated thresholds were dropped. Consequently, only the indicators passing the accuracy test are kept (as illustrated in Table 2). In line with the indications for assessing latent variables when using SEM [88] and given the interchangeable nature of the items measuring a reflective factor, the final number of indicators may be considered suitable as "one or two indicators are often sufficient, but three indicators may occasionally be helpful". (p. 1).

**Table 2.** Constructs and indicators.

| Constructs | Indicators | Cronbach's Alpha | CR | AVE | Sources (Adapted From) |
|---|---|---|---|---|---|
| Strength of team culture (Reflective) | CULT 1. Sharing the same values and goals<br>CULT2. Co-creation of team climate based on trust<br>CULT3. Presence of a strong team spirit | 0.811 | 0.888 | 0.726 | [36,37,50,58] |
| Effectiveness of the communication between leaders and team members (Reflective) | COMM1. Effectiveness of task communication via different channels<br>COMM2. Knowledge sharing for clarifying misunderstandings<br>COMM3. Inspiring members via empathic communication | 0.821 | 0.893 | 0.737 | [5,16,17,19,28,30,31,57] |
| Level of team performance (Reflective) | PERF1. Effectiveness in meeting the established objectives<br>PERF2. Finishing tasks on time<br>PERF3. Good coordination resulting in the usage of less resources | 0.815 | 0.890 | 0.731 | [44,49,56–58] |
| Level of satisfaction with teamwork (Reflective) | SAT1. Enjoying teamwork in virtual teams<br>SAT2. Openness to support teamwork in the virtual environment | 0.895 | 0.950 | 0.905 | [76,77,82,85,88] |

## 4. Results

### 4.1. Measurement Model Evaluation

The statistical scrutiny revealed that the model complies with the GoF criterion of SRMR < 0.08 and the obtained values (SRMR = 0.070 for the saturated model and SRMR = 0.073 for the estimated model), indicating a good model fit.

The measurement model was assessed by means of validity and reliability tests. On this front, composite reliability (CR) was above the 0.7 threshold, whereas the average variance extracted values (AVE) were above the 0.5 threshold for all constructs in line with Hair et al.'s [89] guidelines (as described in Table 2).

The discriminant validity apposite for all constructs was appraised using the Fornell–Larcker and Heterotrait–Monotrait criteria. Consistent with Fornell and Larcker's [90] criterion according to which the AVE values for the latent variables should be higher than the correlation coefficients with the other variables, the measurement model demonstrated accuracy as shown in Table 3. Moreover, the Heterotrait–Monotrait criterion, which establishes a threshold value of 0.90 for structural models with variables that are conceptually similar, was also fulfilled, thus supporting the existence of discriminant validity among the proposed constructs [91].

The level of collinearity of the items was also evaluated. In this respect, all the VIF values were below 3.3 [92], the highest reported value being 2.913. Based on these results, it was concluded that no multicollinearity exists among the indicators.

**Table 3.** Discriminant validity based on Fornell–Larcker criterion.

| | Level of Satisfaction with Teamwork | Effectiveness of the Leader-Team Communication | Strength of Team Culture | Level of Team Performance |
|---|---|---|---|---|
| Level of satisfaction with teamwork | 0.951 | | | |
| Effectiveness of the leader-team communication | 0.630 | 0.858 | | |
| Strength of team culture | 0.608 | 0.743 | 0.852 | |
| Level of team performance | 0.709 | 0.718 | 0.746 | 0.855 |

*4.2. Structural Model Assessment*

For the purpose of testing the five proposed hypotheses, a bootstrapping procedure with 5000 subsamples was carried out. The significant path coefficients are highlighted in Figure 2 and detailed in Table 4.

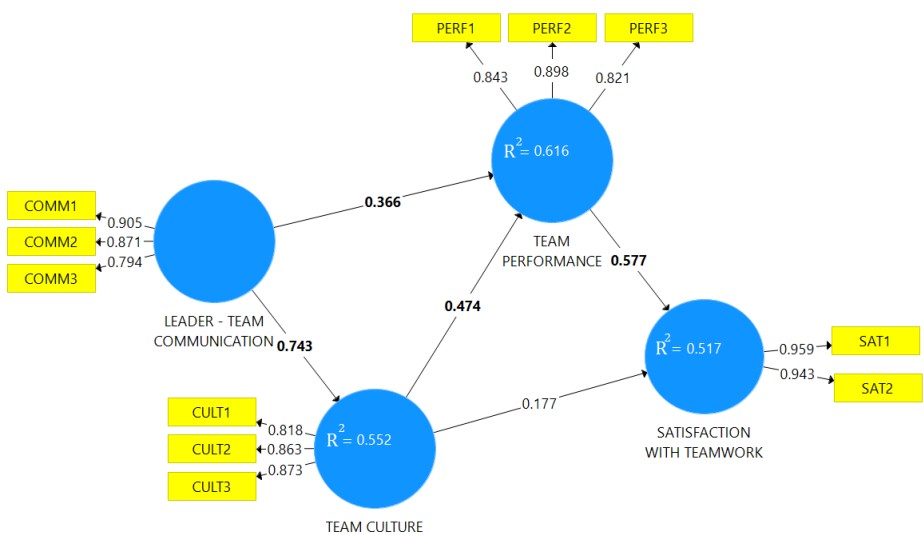

**Figure 2.** Structural model.

**Table 4.** Path coefficients and hypothesis validation.

| Effects | Original Sample | Sample Mean | Standard Deviation | T Statistics | *p* Values | Hypothesis Validation |
|---|---|---|---|---|---|---|
| Effectiveness of leader-team communication -> Level of team performance | 0.366 | 0.365 | 0.068 | 5.415 | 0.000 | H1 supported |
| Effectiveness of leader-team communication -> Strength of team culture | 0.743 | 0.745 | 0.037 | 20.040 | 0.000 | H2 supported |
| Strength of team culture -> Level of team performance | 0.474 | 0.474 | 0.079 | 6.013 | 0.000 | H3 supported |
| Level of team performance -> Level of satisfaction with teamwork | 0.577 | 0.576 | 0.077 | 7.508 | 0.000 | H4 supported |
| Strength of team culture -> Level of satisfaction with teamwork | 0.177 | 0.179 | 0.092 | 1.916 | 0.055 | H5 not supported |

As indicated by the obtained coefficients of determination (*R* square), communication between leaders and team members accounts for 55.2% of the variance in team culture, while both explain 61.6% of the variance in team performance. In their own right, team culture and team performance account for 51.7% of the variance in satisfaction with teamwork.

The unfolding of the structural model assessment brought forward that four of the five inferred relationships between constructs were supported by the empirical investigation.

The testing of the first hypothesis—H1: the effectiveness of the communication between leaders and team members positively influences the level of team performance within virtual teams in the context of the COVID-19 pandemic—indicated a positive influence between the two variables ($\beta = 0.366$; $p < 0.001$), thus H1 was validated. H2 presumed that the effective communication between leaders and team members positively influences a stronger team culture within virtual teams in the context of the COVID-19 pandemic. The obtained values for this relationship ($\beta = 0.743$; $p < 0.001$) validate the formulated hypothesis, supporting a high positive influence of communication effectiveness on the strength of team culture and therefore confirming H2.

Furthermore, the investigation of H3—the strength of team culture positively influences the level of team performance within virtual teams in the context of the COVID-19 pandemic—also indicated a positive influence between the two variables ($\beta = 0.474$; $p < 0.001$), hence supporting H3. The same situation applies for H4—the level of team performance positively influences the level of satisfaction with teamwork within virtual teams in the context of the COVID-19 pandemic—in that the statistical analysis has reported a positive influence between the considered constructs ($\beta = 0.577$; $p < 0.001$). The last hypothesis, H5—the strength of team culture positively influences the level of satisfaction with teamwork within virtual teams in the context of the COVID-19 pandemic—was the only one not supported by the data in the current research context ($p > 0.05$).

Going further, the structural model was analyzed via Cohen's [93] indications for effect sizes, namely small (0.02), medium (0.15) and large (0.35). In this sense, the results highlighted a high effect size in the relationship between the effectiveness of leader-team communication and the strength of team culture ($f^2 = 1.231$), a medium effect size in the relationships between the level of team performance and the level of satisfaction with teamwork ($f^2 = 0.307$), the strength of team culture and the level of team performance ($f^2 = 0.262$) and the effectiveness of leader-team communication and the level of team performance ($f^2 = 0.157$) and a small effect in the relationship between the strength of team culture and the level of satisfaction with teamwork ($f^2 = 0.029$).

The predictive relevance ($Q^2$) was also measured given that the two-item endogenous variable is reflective. Consistent with Fornell and Cha [94], the model has indicated a good predictive relevance for the endogenous construct (i.e., the level of satisfaction with teamwork) ($Q^2 = 0.458$), whereas the variance in satisfaction with teamwork ($R^2 = 0.517$) supported a substantial model.

## 5. Discussion of the Findings

Through the analysis of the acquired data, the proposed theoretical model has been empirically tested. The statistical examination proves positive, significant correlations for all the relationships that have been advanced (H1–H4), except for the association between team culture and satisfaction with teamwork (H5). As shown by the values of the determination coefficients, the effectiveness of team-leader communication strongly influences the strength of the team culture, and there are strong interdependencies between team–leader communication, team culture and team performance, but also a significant effect of the level of team performance on the level of satisfaction with teamwork.

In what concerns the impact of effective communication among the team members on a stronger team culture, these findings confirm that a leader's compelling communication skills are instrumental in inspiring and supporting team members to engage with each other with the view to accomplish their tasks. As a consequence, the team environment becomes a space of solidarity and commitment, where people are reassured that they work together toward a common goal, and everyone's investment will benefit all. Such confidence is even more important in virtual teams, since real face-to-face interactions are missing or are affected by digital media limitations and the perceived physical distance can impede close exchanges. At the same time, advanced information technology options nowadays

offer opportunities for effective management of large, or (internationally) spread out, asynchronous teams, as long as competent and skilled leadership ensures clear information input and expectations, empathic bilateral and group communication, transparent and continuous knowledge sharing with the team members, as well as capable oversight.

These research findings provide evidence that effective communication is paramount to the development of a strong team culture and are in line with previous investigations on the impact of leaders with good communication skills on teamwork [5,28,31,95,96], dialogic communication within teams [25,26], sage managers with effective communication competences [16,19,20,30], trust, team climate [37,50] and team spirit [57,58].

By employing effective communication, leaders can motivate people and persuade them to work together more efficiently to attain the common goals. Furthermore, good communication is the basis for establishing a group ethos with common values and meaningful connections between team members. Apart from the relationship- and culture-building value of effective communication between leaders and team members, cognizant and open communication is of greatest importance for the team performance, as demonstrated also by the results of the current study. This should come as natural, since a clear description of tasks and objectives, as well as good process management, are essential for achieving team and organizational aims. Misunderstandings, blockages and overly complex tasks can be mitigated by sustained sharing of know-how, information exchanges and targeted operational instructions. As a consequence, team members gain confidence in the leader's capabilities to manage teamwork, to overcome obstacles and to lead the team through internal and external pressures.

Work performance in virtual teams can be hampered not only by difficulties specific to traditional teamwork, but further by a sense of disconnection with the teammates. Therefore, effective communication in such environments has been identified in the literature as a main challenge for successful leadership [28], but also as a key remedy, by means of rich channels and consistent exchanges [32]. By promoting the use of the richest digital media, such as video calls and online conferences, that also allow to some extent nonverbal communication, but also, in addition, live chats, phone calls and instant messaging, which are valuable means for virtual teamwork that support swift information sharing and expeditious response to occurring challenges, good leaders can foster engagement, as well as transparent and meaningful communication with the team members.

The empirical evidence proves that relational leadership supported by effective communication impacts team performance, which confirms other authors' findings on leadership and work outcomes [35], engagement in virtual environment [40], effectiveness in achieving objectives [56], timely task completion [43] and efficient use of resources because of improved coordination [44,57,58].

A strong team culture based on trust, participation and transparent communication fosters team performance. In turn, having a clear understanding of the tasks and objectives to be accomplished, confidence that the team leaders offer guidance and support as needed, promoting knowledge sharing and common priorities among team members and relying on a transparent appraisal procedure, leads to an increased sense of professional achievement and personal well-being, which, in the end, determine satisfaction with teamwork. It was rather to be expected that one's contentment with the results of teamwork is influenced by their perception on the recognition of their contribution, interactions with the other team members, and the value attributed to their efforts. It should be noted that individual perceptions are further challenged in virtual teams, as it has been established by recent studies [10] that factors such as inadequate communication, remoteness, workload, etc., can affect performance and welfare. Nevertheless, this research shows that team performance and effective communication significantly impact satisfaction with teamwork in virtual teams. These findings are aligned with the extant literature on satisfaction with virtual teamwork [76] and propension for virtual teamwork [77,82,84,85].

## 6. Conclusions

### 6.1. Summary of the Findings

The assessment of the proposed structural model highlighted that the effective communication between leaders and team members accounts for 55.2% of the variance in the strength of team culture, both explaining 61.6% of the variance in the level of team performance, whereas the strength of team culture and the level of team performance account for 51.7% of the variance in the level of satisfaction with teamwork.

The findings support the advancement of a pertinent conceptual model that mostly relies on significant relationships among constructs. In this respect, four out of five research hypotheses were validated in the context of the current exploration, the highest influences being reported between the effective leader-team communication and the strength of team culture, respectively between the level of team performance and the level of satisfaction with teamwork.

### 6.2. Theoretical and Managerial Contributions and Implications

The study was meant to bring forward meaningful contributions at two levels at least—that is, theoretically and managerially.

In what concerns the theoretical implications, by conducting the empirical survey with members of virtual teams in the COVID-19 pandemic context, the study was intended to capture the state-of-the-art online work environments. Most of the subjects did not have any experience with working online before the outbreak of the pandemic; therefore, their evaluations of key processes unfolding in virtual teams is indicative of the new patterns availed by the new normal. Consequently, the paper looked into topical phenomena and expanded the literature on effective communication, the strength of team culture and the level of team performance in virtual teams while adding new knowledge on the relationship between the level of team performance and satisfaction with teamwork. By doing so, the research advances an integrative model that covers meaningful relationships among constructs which have been either separated investigated or tested beyond the context of COVID-19 and the directly afflicted employees. As previously mentioned, the research sample only comprised respondents who were forced to migrate to the virtual work environment by the outbreak of the pandemic, thus having limited or no prior experience with working online. In this vein, the merit of the paper relies on capturing the genuine influences between constructs in the case of a particular population sample rather than proposing a general structural model for exploring work processes within virtual teams. The investigation thus complements recent research (e.g., [5]) that contrasted traditional versus virtual teams without going further into the underlying relationships among constructs.

Regarding the organizational and managerial implications, the present undertaking aimed to raise awareness on several main issues apposite for the COVID-19-related work environment. First, building effective communication skills proved to be instrumental in managing teamwork and increasing team performance and satisfaction. Second, it was revealed that leaders and managers might be perceived as key factors for fostering team culture, trust and engagement through effective communication strategies. Third, ensuring a positive team climate spurred team performance, which directly and substantively influenced members' satisfaction with teamwork. Conflating these arguments, it may be posited that virtual teams significantly benefit from the development of strong team culture and effective communication and proper importance should be attached to these aspects within the scope of the emergent managerial strategies.

### 6.3. Research Limitations and Future Research Directions

Several research limitations are worth considering and addressing by further empirical studies. On the one hand, when assessing the relationships among different team patterns and processes, the focus was only on virtual teams. In this respect, future research may

envisage testing the same hypotheses in both traditional and virtual teams, or in hybrid work environments with a view to compare the resulting structural models.

On the other hand, the research instrument was based on the ratings of the participants in the study, thus admitting potential subjectivity and bias in statement evaluations. To the best of our knowledge, there have not been developed any particular standardized instruments for measuring the explored constructs in the context of virtual teams; hence, the authors have tried to generate an instrument as derived from previous research dedicated to the work patterns in the virtual environment. Future research may benefit from further testing the constructs using more indicators. The convenience sample prevents the generalization of the findings to the whole population working in virtual teams; therefore, future enterprises might consider extending or refining the sample and to additionally include objective measures for team performance and satisfaction.

Finally, future works may broaden the conceptual area by including other relevant constructs in the model, such as the effectiveness of the communication among team members, intercultural diversity, etc. Additionally, in the case of the relationships that may imply bidirectional effects (e.g., team performance and satisfaction with teamwork in virtual teams), further studies would benefit from theoretically underpinning the reverse influence and provide new insights into the interconnections between the two variables.

**Author Contributions:** Conceptualization, E.-M.V., M.-E.S. and E.D.; methodology, E.-M.V.; software, E.-M.V.; validation, E.-M.V. and E.D.; formal analysis, E.-M.V.; investigation, M.-E.S.; resources, F.V.; data curation, R.-M.S. and F.V.; writing—original draft preparation, E.-M.V., E.D., M.-E.S., R.-M.S. and F.V.; writing—review and editing, E.-M.V.; visualization, R.-M.S.; supervision, E.-M.V.; project administration, E.-M.V.; funding acquisition, E.-M.V. All authors have read and agreed to the published version of the manuscript.

**Funding:** This work was supported by a grant of the Romanian Ministry of Education and Research, CNCS-UEFISCDI, project number PN-III-P1-1.1-TE-2019-1356, within PNCDI III.

**Institutional Review Board Statement:** Not applicable.

**Informed Consent Statement:** Informed consent was obtained from all subjects involved in the study.

**Data Availability Statement:** Not applicable.

**Conflicts of Interest:** The authors declare no conflict of interest.

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
