# Peer review of "Adding Knowledge to Virtual Teams in the New Normal: From Leader-Team Communication towards the Satisfaction with Teamwork"

_sustainability, doi:10.3390/su14116424_

Round 1

Reviewer 1 Report

The article refers to the interesting issues of remote work (in particular virtual teams), which have been recently extremely popular due to the pandemic, but it does not seem to deal with issues related to sustainability or sustainable development, which are the main topic of the journal. Notwithstanding this, I would like to share some comments and concerns:

  • the aim of the work could be more precisely defined; it is rather indicated in a paper what is the subject of the work's interest (lines 14-17 and 88-92). However, it must be admitted that the aim of empirical study is specified precisely (lines 296-298) and the method used fits the goal;
  • one of the biggest shortcomings of the article is that the Authors do not define the basic terms (satisfaction with teamwork, team culture, team performance or, team communication), and do not precise what aspects of them are exactly an object of measurement, e.g. in case of leader-team communication – it can be effectiveness, quality, intensity, satisfaction from communication, communication responsiveness, etc. or in case of team culture - culture type, strength, dimensions, etc. This is of great importance because the hypotheses are formulated very generally, eg H2: 'The communication between leaders and team members positively influences team culture’, which raises questions: Does the mere fact of communication has a positive effect on team culture?; What aspect / dimension of the team culture influences it? etc. I suggest reducing both the superficiality of the description and the level of generalization in this case;
  • survey items used to build research constructs look rather random; Authors building scales refer to the literature sources, however they do not justify why the particular items should be included in the evaluation, what they measure, or why couldn’t we use scales proposed by other literature sources;
  • the research topic is not new; all pointed hypothesis are rather well grounded in theory and supported by quoted research results, some of them also in the context of virtual teams; hence it arises the question what is the Authors contribution to the state-of-art? Authors, going to enrich the knowledge, concentrate on the relation between team performance and satisfaction with teamwork in virtual teams. The results obtained clearly indicate that team performance strongly, positively influences satisfaction with teamwork, and that would be interesting especially in the context of the previous statement that the ‘straightforward causal relationship between job satisfaction and performance has not been proven yet’ (lines 223-224). However, this issue, one employee level, is a subject of psychologists research from at least last 50 years. There is no objection that these variables are positively related, however not as much as if it resulted from our intuitive feeling. Also, the direction of their relationship is not quite clear. There are research stating that satisfaction with work is influenced by performance, however, there are also such pointing that the opposite direction of correlation is also possible. Therefore, it is doubtful whether the results obtained are not strongly dependent on the chosen method of measuring the research variables.

Reviewer 2 Report

The paper is quite interesting and well written. The basic research question is sound and interesting. The COVID-19 impact on working groups and satisfaction is profoundly deep and re-established working manners and overall status.

The paper deals with knowledge, communication, satisfaction, team culture and performance of teams in a satisfactory way, but I would suggest some improvements as the following

  1. Regarding the research instrument (questionnaire) I feel that the constructs creating the variables are insufficient. For instant Satisfaction could not be sufficiently explained with only two constructs. The rest of the variables contain 3 variables each, which is also - according to my point of view unable to explain clearly the variables. 
  2. Moreover, I would suggest working more on the statistical analysis. Since you are implementing the SEM methodology, it would be quite interesting to give us a more detailed picture of the produced model.  
  3. Please also state the Cronbach's alpha indicator, just to make sure of the internal cohesion and reliability of the questionnaire. 
  4. The last issue is the (low) sample of 175 questionnaires. It could be argued that 10 constructs shared amongst variables, might be enough but since we are working on a SEM model it would be quite preferable to increase the sample (and reach 300 fulfilled questionnaires).

In general lines, the paper is (as I have mentioned) well written, but it could be improved. 

Reviewer 3 Report

Dear authors, I consider the topic is interesting. However, the document can be improved.

I propose the following changes:

1. In the hypotheses, define their scope, for example, in H1 write:
H1: Communication between leaders and team members positively influences team performance does telework (or virtual work) in times of the Covid-19 pandemic.

Do the same for all hypotheses.

2. It is necessary that the authors make reference to Figure 1 in the text. For instance: “Corroborating all the inferred relationships, research model in shown in Figure 1 was proposed.

3. The authors include the results in the methodology section (as shown in Table 1, Table 2 and Table 3). The methodology only has to show how the research was conducted, it must not include results. 

Lines 315-320 include results, but they are in mehtodology

Authors must explain more about the Fornell- 323 Larcker and Heterotrait-Monotrait criteria. What does it says?

Methodology ust be in present tense, not in past tense.

4. Lines 330-332: Here, the authors present results, but they are in methodology.

5. In Figure 2, for the numbers in the cycles, I suggest writing R2 = The number, so the readers can easily know what those numbers mean.

6. References 11, 15, 20, 61, 86: The years of publication are necessary.

7. References 36, 37, 41, 66, 89 are obsolete. Include more recent references

I hope these comments help to improve the quality of the manuscript.

Round 2

Reviewer 2 Report

My previous basic observation was about the few questions (items) that constitute the essential research variables. I believe that the reliability of the research could be questioned, but on the other hand, the authors tried to meet my suggestions. Under this condition and the paper is well written and clear to the final outcomes and conclusions.

Author Response

Dear reviewer,

thank you for your kind comments!

Following your first-round suggestions, we have revisited the literature on the minimum required items to measure a construct and we found the explanations already included in the paper. 

Thank you for accepting our argumentation and for all your thorough inputs!

Reviewer 3 Report

The manuscript has been improved.

However, the following references have a dot (.) just before the last name of the first author:

36. . Hofstede, G. Dimensionalizing Cultures: The Hofstede Model in Context. Onl. Read. Psy. Cult. 2011, 2, 8, http://dx.doi.org/10.9707/2307-0919.1014.

41. . Purvanova, R.K.; Kenda, R. The impact of virtuality on team effectiveness in organizational and non-organizational teams: A meta-analysis. App. Psy. 2021, https://doi.org/10.1111/apps.12348.

Delete these dots (.).

Author Response

Dear reviewer,

thank you for all your support along the way!

We have deleted the two dots.